# GRAPHSEARCH: AGENTIC SEARCH-AUGMENTED REASONING FOR ZERO-SHOT GRAPH LEARNING

## ABSTRACT

Recent advances in search-augmented large reasoning models (LRMs) enable the retrieval of external knowledge to reduce hallucinations in multistep reasoning. However, their ability to operate on graph-structured data, prevalent in domains such as e-commerce, social networks, and scientific citations, remains underexplored. Unlike plain text corpora, graphs encode rich topological signals that connect related entities and can serve as valuable priors for retrieval, enabling more targeted search and improved reasoning efficiency. Yet, effectively leveraging such structure poses unique challenges, including the difficulty of generating graph-expressive queries and ensuring reliable retrieval that balances structural and semantic relevance. To address this gap, we introduce **GraphSearch**, the first framework that extends search-augmented reasoning to graph learning, enabling zero-shot graph learning without task-specific fine-tuning. GraphSearch combines a *Graph-aware Query Planner*, which disentangles search space (e.g., 1-hop, multi-hop, or global neighbors) from semantic queries, with a *Graph-aware Retriever*, which constructs candidate sets based on topology and ranks them using a hybrid scoring function. We further instantiate two traversal modes: GraphSearch-R, which recursively expands neighborhoods hop by hop, and GraphSearch-F, which flexibly retrieves across local and global neighborhoods without hop constraints. Extensive experiments across diverse benchmarks show that GraphSearch achieves competitive or even superior performance compared to supervised graph learning methods, setting state-of-the-art results in zero-shot node classification and link prediction. These findings position GraphSearch as a flexible and generalizable paradigm for agentic reasoning over graphs.

## 1 INTRODUCTION

Large Reasoning Models (LRMs) have recently shown strong capabilities in solving complex problems through extended, stepwise reasoning (e.g., OpenAI et al. (2024); Qwen (2025); DeepSeek-AI et al. (2025)). Yet, this deliberate reasoning process often suffers from knowledge insufficiency, where missing or outdated facts can propagate errors and lead to hallucinations. To address this limitation, recent work integrates agentic search into reasoning (e.g, Yao et al. (2023); Li et al. (2025); Jin et al. (2025); Song et al. (2025a); Chen et al. (2025)), enabling LRMs to dynamically retrieve and incorporate up-to-date external knowledge during their reasoning rollouts. By tightly coupling retrieval with reasoning, these frameworks substantially improve factual grounding and reliability, marking an important step toward more trustworthy large-scale reasoning.

However, their ability to operate on graph-structured data, which is practical and pervasive in real-world domains such as e-commerce, social networks, and scientific citations, remains largely unexplored. Unlike unstructured text, graphs encode explicit topological signals that link related entities and serve as essential priors for core graph learning tasks such as node classification and link prediction. For example, in citation networks Hu et al. (2020), classifying a paper depends not only on its own metadata but also on the metadata of the papers it cites. Yet graph learning models transfer poorly across graphs and tasks, and each new graph typically requires collecting labeled data and retraining, leading to high annotation costs and limiting scalability. This makes zero-shot generalization highly desirable Sun et al. (2025b); Jiang et al. (2024). Standard search-augmented LRMs, however, treat each node (i.e., query) in isolation and fail to leverage these structural relations, leading to suboptimal performance on graph-learning tasks, as shown in Figure 1.

Enabling structure-aware agentic search over graphs is fundamentally challenging. First, **query expressivity** is limited: natural language queries cannot specify graph-structured requirements such as hop-based neighborhoods. Even if "neighbors" are mentioned, retrievers typically interpret them as keywords rather than traversal instructions, overlooking structural signals critical for reasoning. Second, **retrieval reliability** is difficult to guarantee: generated queries are often noisy or variable, and effective retrieval requires both (i) candidate sets grounded in graph topology and (ii) ranking functions that balance semantic relevance with structural context.

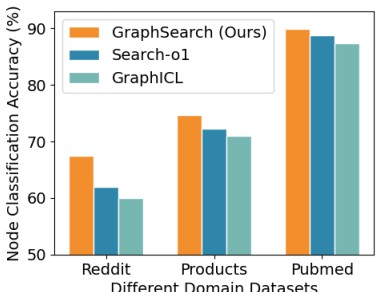

Figure 1: **Effect of structure awareness and dynamic search.** Search-o1 is structure-agnostic, while GraphICL relies on static neighbor injection without dynamic exploration. GRAPHSEARCH introduces an agentic and structure-aware search rollouts that enable more effective graph learning.

Existing attempts Sun et al. (2025b); Huang et al. (2024); Wang et al. (2025) to adapt LRMs for graphs often rely on *static context injection*, where the anchor node and its neighbors are directly appended to the input prompt. This approach faces a fundamental trade-off: including all neighbors introduces noise and quickly exceeds context length, while selecting a subset requires heuristics that rarely generalize across graphs or tasks, especially in zero-shot settings. As shown in Figure 1, both the structure-agnostic search and static neighbor injection without dynamic exploration yield suboptimal performance, underscoring the need for an agentic, structure-aware approach to graph reasoning tasks. Another line of work explores retrieval-augmented generation on constructed semantic graphs, such as GraphRAG Edge et al. (2025). However, these systems operate on document-level co-occurrence graphs rather than native graph topology and are designed for factoid QA rather than prediction-oriented graph learning.

To address these challenges, we propose **GraphSearch**, a novel framework that extends agentic search-augmented reasoning to graph-structured data, enabling zero-shot graph learning without task-specific fine-tuning. GraphSearch is built around two key components: A *Graph-aware Query Planner*, which disentangles the search space (e.g., 1-hop, multi-hop, or global neighborhoods) from the semantic query, allowing LRMs to issue structured, graph-expressive search instructions on the fly. A *Graph-aware Retriever*, which (i) constructs candidate sets based on graph topology, (ii) ranks them using a hybrid scoring function that balances anchor–candidate similarity and candidate–query similarity, and (iii) instantiates two traversal modes: GraphSearch-R, which recursively expands hop-1 neighborhoods step by step like message passing, and GraphSearch-F, which flexibly retrieves from local and global neighborhoods without hop constraints. Our graph-aware agentic framework can also be extended into a learnable mechanism when training data is available. We show that GraphSearch can be further optimized via reinforcement learning (RL) using GRPO DeepSeek-AI et al. (2025) and a simple reward design, covering format correctness, answer accuracy, and search usefulness, demonstrating that the framework naturally supports learning-based refinement.

In summary, our key **contributions** are summarized below:

★ We study search-augmented reasoning in graph-structured domains and propose GraphSearch, the first framework that extends search-augmented reasoning to graph learning tasks through graph-guided, multi-step agentic rollouts, enabling zero-shot prediction without any fine-tuning.

★ We introduce two core components: a *Graph-Aware Query Planner* that enables expressive search space control, and a *Graph-Aware Retriever* that combines topology-grounded candidate construction with hybrid semantic–structural ranking. The retriever is instantiated in two complementary modes: GraphSearch-R (recursive, hop-by-hop) and GraphSearch-F (flat, global access), supporting both localized and long-range reasoning.

★ We validate GraphSearch on six benchmark datasets for node classification and link prediction across multiple graph domains. The empirical results demonstrate its effectiveness over state-of-the-art graph learning methods and its superior efficiency compared to standard agentic search-augmented LRM approaches.

★ We further show that GraphSearch naturally accommodates learnable extensions: a lightweight GRPO-based RL fine-tuning yields stable performance gains over its training-free variant.

## 2 RELATED WORK

Our work is closely related to the following two directions.

**Search-augmented Reasoning Models.** Large reasoning models (LRMs) often face knowledge gaps, which have motivated search-augmented methods that integrate external retrieval into step-wise reasoning. For example, Search-o1 Li et al. (2025) augments LRMs with an agentic retrieval-augmented generation (RAG) mechanism in the reasoning process, while subsequent efforts such as Search-R1 Jin et al. (2025), R1-Searcher Song et al. (2025a), R1-Searcher++ Song et al. (2025b) and ZeroSearch Sun et al. (2025a) further optimize query planning and retrieval strategies to improve reasoning performance. However, these approaches mainly operate over unstructured text corpora and largely ignore the intrinsic structural information in graph data, where reasoning requires exploiting graph topology and relational dependencies.

**RAG with Graph.** GraphRAG-style methods Edge et al. (2025); Gutiérrez et al. (2025); Guo et al. (2025) build chunk-level semantic graphs from text to support multi-hop, document-oriented factoid QA. Because these graphs are derived from text chunks rather than the native relational topology of a real graph dataset (e.g., citation links, co-purchase edges, or social connections), they do not preserve structural priors crucial for prediction-oriented graph learning (e.g., neighborhoods, connectivity). GraphCoT Jin et al. (2024) uses native graph links but is tailored for factoid QA with fixed function calls and no graph-aware query generation, and is not designed for graph learning tasks. Within graph learning, retrieval-augmented methods like RAGraph Jiang et al. (2024) provide a toy-graph retrieval step, but still rely on pre-trained GNNs. Overall, these approaches target text-centric QA or remain tied to GNN pipelines, leaving open how to perform search-augmented reasoning directly on graph-structured data in a zero-shot manner.

**LLM for Graph Learning.** Inspired by the rapid advancement of LLMs, their potential for solving graph data tasks has attracted growing research interest. Existing efforts typically train LLMs for graph learning either by introducing a projector to align graph embeddings with the LLM embedding space (e.g, Zhang et al. (2024); Liu et al. (2024); Chen et al. (2024); Zhang et al. (2024); Wang et al. (2024)), or by applying supervised fine-tuning (SFT) to adapt the LLM to downstream tasks Tang et al. (2024). While effective, these training-based paradigms often incur substantial computational cost and depend on task-specific labels, which are scarce in many real-world scenarios You et al. (2021). To alleviate these limitations, prompt-based methods leverage in-context learning to directly inject structural information into task prompts Sun et al. (2025b); Hu et al. (2025); Huang et al. (2024); Wang et al. (2025), guiding LLM predictions without retraining. Despite their lightweight design, such approaches all rely on statically predefined structural information (e.g., neighbor sets), which introduces scalability challenges and exacerbates context length constraints.

## 3 METHODOLOGY

This section introduces GRAPHSEARCH, an agentic search-augmented framework for reasoning over graph data. We first describe the overall agentic rollout, then detail the two core innovations: the *Graph-Aware Query Planner* (query expressivity) and the *Graph-Aware Retriever* (retrieval reliability), including two traversal variants (*GraphSearch-R/F*).

### 3.1 PROBLEM SETUP AND AGENTIC ROLLOUTS

We consider *node classification* and *link prediction* tasks on a graph $\mathcal{G} = (\mathcal{V}, \mathcal{E})$ with node attributes $\text{Attr}_i$ for $v_i \in \mathcal{V}$. For node classification, given an anchor node $v_i$, the goal is to predict its label; for link prediction, given a pair $(v_i, v_j)$, the goal is to infer the existence of an edge. Given a task instruction $I$, the graph $\mathcal{G}$, and the anchor input $\mathcal{N}_{\text{anch}}$ (a node or node pair), an LRM produces a reasoning trajectory $\mathcal{R}$ interleaved with searches, and a final answer $\mathcal{A}$. We denote this process by a probabilistic model $F(\cdot)$ over reasoning and answer sequences conditioned on the inputs:

$$F(\mathcal{R}, \mathcal{A} \mid I, \mathcal{N}_{\text{anch}}, \mathcal{G}) = \underbrace{\prod_{t=1}^{\mathcal{T}_{\mathcal{R}}} F(\mathcal{R}_t \mid \mathcal{R}_{<t}, I, \mathcal{N}_{\text{anch}}, \mathcal{G}_{<t})}_{\text{Reasoning Process}} \cdot \underbrace{\prod_{t=1}^{\mathcal{T}_{\mathcal{A}}} F(\mathcal{A}_t \mid \mathcal{A}_{<t}, \mathcal{R}, \mathcal{N}_{\text{anch}}, \mathcal{G}_{<t})}_{\text{Answer Generation}}. \quad (1)$$

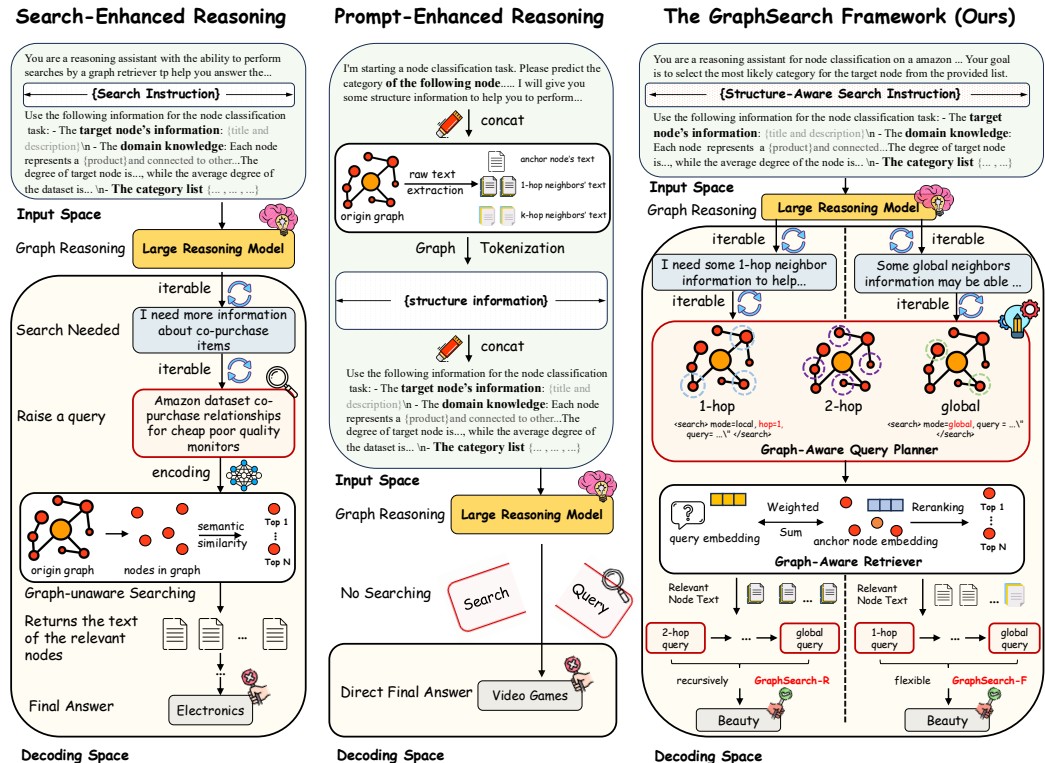

Figure 2: Comparison of graph reasoning approaches: (i) *Search-Enhanced Reasoning* (left) dynamically retrieves external information based on query–semantic similarity; (ii) *Prompt-Enhanced Reasoning* (middle) statically injects pre-defined structural cues into the prompt; (iii) *GraphSearch* (right) enables dynamic retrieval of structural information through a graph-aware query planner and retriever.

where $\mathcal{T}_{\mathcal{R}}$ and $\mathcal{T}_{\mathcal{A}}$ are the lengths of the reasoning and answer sequences, $\mathcal{R}_t$ is the reasoning state at step $t$ with history $\mathcal{R}_{<t}$, and $\mathcal{G}_{<t}$ denotes the structural information retrieved from $\mathcal{G}$ up to step $t$.

**Agentic Structural Search.** The agentic rollout alternates between *think* and *search* steps (Eqs. 1, 2), delimited by `<think>...</think>` and `<search>...</search>`. Content inside `<search>...</search>` is parsed by the *Graph-Aware Query Planner* $P(\cdot)$ into a structured instruction $\mathcal{Q}_t$, then forwarded to the *Graph-Aware Retriever* $S(\cdot)$. The returned evidence $\mathcal{G}_t$ is injected via `<information>...</information>` and conditions subsequent reasoning steps. Once sufficient evidence has accumulated, the model emits the final solution within `<answer>...</answer>`.

$$\underbrace{F(\mathcal{R} \mid I, \mathcal{N}_{\text{anch}}, \mathcal{G})}_{\text{Reasoning Process}} = \prod_{t=1}^{\mathcal{T}_{\mathcal{R}}} \Big[ \underbrace{F(\mathcal{R}_t \mid \mathcal{R}_{<t}, I, \mathcal{N}_{\text{anch}}, \mathcal{G}_{<t})}_{\texttt{<think>}} \cdot \underbrace{P(\mathcal{Q}_t \mid \mathcal{R}_{<t}, I, \mathcal{N}_{\text{anch}}, \mathcal{G}_{<t})}_{\texttt{<search>}} \cdot \underbrace{S(\mathcal{G}_t \mid \mathcal{Q}_t, \mathcal{G})}_{\texttt{<information>}} \Big]. \quad (2)$$

Here, $P(\cdot)$ emits a structured instruction $\mathcal{Q}_t$ (or $\varnothing$ when no search is issued), and $S(\cdot)$ maps $\mathcal{Q}_t$ to an evidence block $\mathcal{G}_t \subseteq \mathcal{G}$ (or returns $\varnothing$ when $\mathcal{Q}_t = \varnothing$). We denote $\mathcal{G}_{<t} = \{\mathcal{G}_1, \ldots, \mathcal{G}_{t-1}\}$ as previously retrieved evidence.

♠ *Core Idea.* Effective graph reasoning hinges on *(i)* a query planner $P(\cdot)$ that yields expressive queries $\mathcal{Q}_t$ and *(ii)* a graph-aware retriever $S(\cdot)$ that grounds candidate selection in topology while ranking semantically relevant nodes.

## 3.2 GRAPH-AWARE QUERY PLANNER

Natural language cannot reliably encode structural constraints (e.g., "2-hop neighbors"), often collapsing them into simple keyword matches. This leads to shallow retrieval, loss of topological

context, and underutilization of graph structure. We argue that effective queries over graph data should be both graph-expressive and executable within a requested structural search space.

To this end, we introduce a graph-aware query planner that guides the LRM to generate queries that are semantically expressive, structurally grounded, and executable with explicit structural scope.

### 3.2.1 GRAPH-SPECIFIC INSTRUCTIONS

We provide the LRM with graph-aware instructions that embed task-specific structural cues, guiding it to generate queries that extend beyond semantic content by explicitly specifying the structural information required for retrieval:

$$\mathcal{I} = \big(\mathcal{I}_{\text{task}},\ \mathcal{I}_{\text{format}},\ \mathcal{I}_{\text{policy}}\big),$$

where $\mathcal{I}_{\text{task}}$ states the task objectives (e.g., node classification or link prediction), $\mathcal{I}_{\text{format}}$ specifies the structured query fields to emit inside `<search>...</search>`, and $\mathcal{I}_{\text{policy}}$ instructs the LRM on guides when and what to search, given the current reasoning state. Detailed instructions are provided in Appendix A.3. Formally, the planner generates the search query at each step $t$ with the guidance from $I$:

$$P(\mathcal{Q}_t \mid \mathcal{R}_{<t}, I, \mathcal{N}_{\text{anch}}, \mathcal{G}_{<t}) = \prod_{i=1}^{\mathcal{T}_{\mathcal{Q}_t}} F(q_t^i \mid q_t^{<i}, \mathcal{R}_{<t}, \mathcal{I}_{\text{task}}, \mathcal{I}_{\text{format}}, \mathcal{I}_{\text{policy}}, \mathcal{N}_{\text{anch}}, \mathcal{G}_{<t}) \quad (3)$$

where $q_t^i$ is the $i$-th token of the query $\mathbf{Q}_t$ at step t and $q_t^{<i}$ represents its preceding tokens.

### 3.2.2 GRAPH-ORIENTED SEARCH QUERIES.

We disentangle structural specifications from semantic content to make queries graph-expressive, explicitly separating where to search from what to search for. At step $t$, the planner outputs a query $\mathcal{Q}_t$ in the following form: $\mathcal{Q}_t = (\mathcal{S}_t, \mathcal{C}_t)$, where $\mathcal{C}_t$ is a semantic query derived from the reasoning context and anchor attributes, and $\mathcal{S}_t$ specifies the structural search space.

To enable flexible retrieval under both dense and sparse graph structures, we design three search-space modes: $\mathcal{S}_t \in \{local, global, attribute\}$. The *local* mode restricts retrieval to the hop-$k$ neighborhood of the anchor, capturing fine-grained structural context, while the *global* mode expands the search to the entire graph, allowing retrieval beyond local neighborhoods. The *attribute* mode complements these topology-based scopes by retrieving nodes through the attribute similarity of the anchor node, providing coverage when structural signals are weak or sparse.

---

**Graph-Aware Query Planner**

The query lacks the ability to specify the search scope:

```
Papers citing Outperforming the Gibbs sampler ...
```

Our graph-oriented query disentangles structure and semantic information:

```
mode=(local, hop=1), query="Markov chain sampling Gibbs sampler"
```

---

♦ **Remark of Graph-Aware Query Planner.** Disentangling structural and semantic signals provides LRMs with fine-grained control over the retrieval candidate scope (topology), independent of semantic content, thereby enabling adaptive, structure-aware reasoning rollouts.

### 3.3 GRAPH-AWARE RETRIEVER

To handle dynamic queries that integrate both structural proximity and semantic relevance, we propose a two-stage retriever for graph data. It first constructs a topology-grounded candidate set, then ranks the candidates with a hybrid score that mitigates query noise by leveraging anchor attributes.

Moreover, building on message aggregation paradigms, we introduce two traversal modes. *GraphSearch-R* recursively expands hop by hop to form a complete subgraph, whereas *GraphSearch-F* flexibly integrates multiple structural scopes without sequential expansion.

### 3.3.1 CANDIDATE CONSTRUCTION VIA SPACE SIGNAL.

The retrieval candidate set is determined by the search space signal $\mathcal{S}_t$ in the query. We first define three candidate neighbor sets with respect to the anchor node $\mathcal{N}_{\text{anch}}$.

- Local Neighbor Set $\mathcal{N}_{\text{anch}}^{\text{local}(h)}$: Nodes within $h$ hops of the anchor node $\mathcal{N}_{\text{anch}}$, capturing its immediate structural neighborhood.

- Global Neighbor Set $\mathcal{N}_{\text{anch}}^{\text{global}}$: Nodes retrieved via personalized PageRank (PPR) Haveliwala (2003), capturing long-range dependencies and globally relevant context beyond the local ones.

- Attribute Neighbor Set $\mathcal{N}_{\text{anch}}^{\text{attribute}}$: Nodes retrieved based on attribute-level similarity to the anchor, providing semantic relevance when structural cues are weak or sparse.

Depending on the input signal $\mathcal{S}_t$, these sets can be selected individually or combined to form the final retrieval space. Formally, the final constructed candidate set for $\mathcal{N}_{\text{anch}}$ is:

$$\mathcal{N}_{\text{anch}}^{\text{candidate}} = \delta_{\text{local}} \cdot \mathcal{N}_{\text{anch}}^{\text{local}(h)} \ \cup \ \delta_{\text{global}} \cdot \mathcal{N}_{\text{anch}}^{\text{global}} \ \cup \ \delta_{\text{attribute}} \cdot \mathcal{N}_{\text{anch}}^{\text{attribute}}, \tag{4}$$

where $\delta_{\text{local}}, \delta_{\text{global}}, \delta_{\text{semantic}} \in \{0, 1\}$ are activation coefficients controlled by the field in search space signal $\mathcal{S}_t$. If $\delta_x = 1$, the corresponding neighbor set is included; if $\delta_x = 0$, it is ignored.

For link prediction with an anchor pair in $\mathcal{N}_{\text{anch}}$, the LRM is instructed to select one node as the query anchor for each request in the query. See Appendix A.3 for details.

### 3.3.2 HYBRID RANKING VIA QUERY AND ATTRIBUTES

Feeding back the entire candidate set would overwhelm the model with redundancy and noise. Instead, we rank candidates and return only the top-$k$ most relevant neighbors to the LRM.

We design a weighted semantic similarity function that combines the query text with the anchor node's attributes, as formalized in Eq. 6, to provide more reliable ranking in graph settings where queries may be unstable or noisy.

$$\text{Score}_j = \alpha \cdot \text{CosSim}\big(\phi(Attr_j), \phi(Attr_{\text{anch}})\big) + (1 - \alpha) \cdot \text{CosSim}\big(\phi(Attr_j), \phi(C_t)\big), \tag{5}$$

$$\mathcal{G}_t = \text{TopK}_{v_j \in \mathcal{N}_{\text{anch}}^{\text{candidate}}}\big(\text{Score}_j, k\big). \tag{6}$$

Here, $\text{CosSim}(\cdot, \cdot)$ denotes cosine similarity between embeddings, and $\phi(\cdot)$ is the encoder that maps textual information into embeddings. $\text{Score}_j$ represents the relevance score of candidate neighbor $v_j$, determined by its similarity to both the anchor node's attributes $Attr_{\text{anch}}$ and the query representation $C_t$. Finally, $\mathcal{G}_t$ is obtained by selecting the top-$k$ candidates from $\mathcal{N}_{\text{anch}}^{\text{candidate}}$, whose attributes are injected back into `<information>...</information>` for reasoning.

$\alpha \in [0, 1]$ is a weighting coefficient that balances the contributions of the anchor node's attributes and the query contents within the structurally filtered candidate set. When $\alpha \to 1$, ranking relies primarily on anchor–based similarity within the structural scope, while when $\alpha \to 0$, it depends solely on query–based semantic similarity.

### 3.3.3 TRAVERSAL VARIANTS: GRAPHSEARCH-R AND GRAPHSEARCH-F

Building on message-passing intuitions, we instantiate two traversal policies inside the retriever to support structure-aware search. *GraphSearch-R* performs recursive, hop-by-hop expansion from the anchor, yielding localized, path-consistent subgraphs. In contrast, *GraphSearch-F* enables flexible retrieval by aggregating candidates from planner-specified scopes.

- *GraphSearch-R*: At each step, restrict candidates to the current anchor's 1-hop neighbors, so the $h$-th search corresponds to the $h$-hop neighborhood. This rollout expands one hop per step—akin to message passing—yielding localized, path-consistent exploration. For sparsely connected nodes, global neighbors are incorporated as needed to meet the retrieval size.

---

**Algorithm 1:** GraphSearch Inference

---

**Input:** Reasoning model $\mathcal{M}$; Instruction $I$; Search function SEARCH; Traversal version v ;
Graph $G$; Anchor node $N$.
**Output:** Reasoning path $R$ and Final answer $A$.

$R \leftarrow \emptyset; A \leftarrow \varnothing ; P \leftarrow \texttt{BuildInitialPrompt}(I, N)$
**while** *true* **do**
    $R \leftarrow R \,\|\, \mathcal{M}.\texttt{Generate}(P, \texttt{until} = \{\texttt{</search>},\texttt{EOS}\})$
    **if** *P ends with* `</search>` **then**
        $Q \leftarrow \texttt{ParseQuery}(R)$   // Extract query preceding the latest `</search>`
        $Info \leftarrow \texttt{Search}(N, Q, \text{v})$     // Retrieve relevant context from graph $G$
        $P \leftarrow P \,\|\, \texttt{<information>}Info\texttt{/<information>}$
        **continue**
    **else if** *P ends with* EOS **then**
        $A \leftarrow \texttt{ParseAnswer}(P)$      // Extract from `<answer> </answer>` block
        **break**
**return** $R, A$

---

- *GraphSearch-F*: Each search step may use *local* / *global* / *attribute* as specified by $S_t$ from the planner, allowing local and global access without hop-wise constraints and enabling broader structural coverage in fewer steps. Note that when *local* is selected, the specific hop number should also be indicated.

♦ **Remark of Graph-Aware Retriever.** By combining candidate construction with hybrid semantic ranking, the two-stage design equips retrieval with both structural sensitivity and semantic alignment, thereby supporting adaptive zero-shot graph reasoning.

### 3.4 GRAPHSEARCH INFERENCE PROCESS

The overall inference procedure of GRAPHSEARCH interleaves reasoning, query planning, and graph retrieval until a final answer is produced, as summarized in the Algorithm 1.

This design provides two key properties: i) Dynamic Retrieval. The LRM autonomously decides when and how to retrieve external information, avoiding fixed scopes and costly context enumeration. This flexibility enables adaptive use of graph-structured signals, allowing the model to better exploit relational patterns in the data. ii) Efficiency. By explicitly disentangling where to search (topology) from what to search (semantics), candidate sets are restricted to local or global neighborhoods rather than the entire graph. This significantly narrows the search space and ensures that ranking is performed only on structurally meaningful subsets, thereby improving inference efficiency compared to brute-force baselines that retrieve over the whole graph.

## 4 EXPERIMENTS

In this section, we conduct experiments to address the following research questions: **RQ1**: Can *GraphSearch* achieve state-of-the-art results on various graph learning tasks across benchmarks from diverse domains? **RQ2**: What are the individual contributions of key components, i.e., graph-aware query planner and graph-aware retriever, to the overall effectiveness of *GraphSearch*? **RQ3**: Can *GraphSearch* be extended to a learnable setting, and does such adaptation further improve its performance? **RQ4**: How efficient is *GraphSearch* compared to standard agentic search-augmented baselines?

### 4.1 EXPERIMENT SETUP

**Datasets.** We evaluate *GraphSearch* on six benchmark datasets spanning three domains and two graph learning tasks (node classification and link prediction). In the **E-commerce domain**, we use OGB-Products He et al. (2023), Amazon-Sports-Fitness Shchur et al. (2018), and Amazon-Computers. In the **Citation domain**, we adopt Cora McCallum et al. (2000) and PubMed Sen et al.

(2008). In the **Social Network domain**, we employ Reddit Yan et al. (2024). Further details are provided in Appendix A.2.1.

**Baselines.** We evaluate three categories of zero-shot baselines. (1) **Direct Reasoning**: These models directly solve the task without search, relying solely on their intrinsic reasoning ability. We use the open-source models Qwen2.5-32B-Instruct (Qwen-32B) Team (2024) and Qwen2.5-72B-Instruct-AWQ (Qwen-72B); (2) **Search-Augmented LRMs**: These models leverage both internal knowledge and externally retrieved information for reasoning, but they cannot perform structured graph search. We include Search-o1 equipped with Qwen-32B and Qwen-72B; (3) **In-Context Learning Graph LLMs**: We incorporate GraphICL, which explores a wide range of template combinations, is specifically designed for graph leanrning but lacks dynamic search capabilities. For reference, we also compare against training-based approaches; (4) **Graph Learning Methods**: GNN-based models (GCN Kipf & Welling (2017), SAGE Hamilton et al. (2017), RevGAT Li et al. (2021), MLP) and LLM-based methods (LLaGA Chen et al. (2024), GraphGPT Tang et al. (2024), GraphTranslator Zhang et al. (2024), GraphPrompter Liu et al. (2024)); (5) GraphRAG Methods: We include Graph-CoT Jin et al. (2024), which also leverages native graph structure for reasoning, as well as classical RAG-based approaches such as GraphRAG Edge et al. (2025) and HippoRAG2 Gutiérrez et al. (2025) for comparison. All implementation details are provided in the Appendix A.2.

## 4.2 Main Results on Zero-shot Graph Learning (RQ1)

To answer **RQ1**, we conduct node classification experiments on six datasets to compare GraphSearch with the baselines, with the main results reported in Table 1.

Table 1: Zero-shot node classification accuracy (%) of GraphSearch and baselines on six datasets. **The Best** and the second best performance within each LRM backbone and dataset are highlighted. GraphICL-S1 and -S2 denote the first- and second-best GraphICL variants. Avg.Rank denotes the average ranking of each method across different datasets under the same backbone.

| Model | Products | Sports | Computers | Reddit | PubMed | Cora | Avg.Rank |
|---|---|---|---|---|---|---|---|
| **Qwen2.5-32B-Instruct** | | | | | | | |
| Graph-CoT | 64.6 | 57.2 | 62.6 | 56.1 | 70.5 | 57.9 | 6.5 |
| Chain-of-Thought | 70.1 | 52.9 | 57.3 | 58.8 | 86.6 | 65.7 | 5.7 |
| GraphICL-S1 | 71.2 | 59.6 | 66.5 | 66.2 | **91.0** | **68.5** | 2.2 |
| GraphICL-S2 | 68.4 | 57.5 | 65.6 | 65.7 | 86.3 | 67.9 | 4.2 |
| Search-o1 | 68.1 | 59.7 | 60.9 | 62.0 | 89.1 | 63.5 | 5.0 |
| GraphSearch-R | 70.8 | **61.9** | 68.1 | 63.5 | 90.4 | 65.9 | 2.7 |
| GraphSearch-F | **71.7** | 59.8 | **69.9** | **67.4** | 89.8 | 67.5 | **1.8** |
| **Qwen2.5-72B-Instruct** | | | | | | | |
| Chain-of-Thought | 66.4 | 58.9 | 60.7 | 59.2 | 80.3 | 65.5 | 5.7 |
| GraphICL-S1 | 71.0 | 69.6 | 70.9 | **70.0** | 87.4 | **70.0** | 2.3 |
| GraphICL-S2 | 69.0 | 68.3 | 70.2 | 68.3 | 79.3 | 66.5 | 4.3 |
| Search-o1 | 72.3 | 69.1 | 60.4 | 60.3 | 88.8 | 66.2 | 4.3 |
| GraphSearch-R | **74.6** | **72.0** | 69.6 | 66.6 | 89.9 | 68.8 | 2.3 |
| GraphSearch-F | 73.2 | 69.2 | **71.0** | 68.7 | **90.0** | 67.2 | **2.0** |

We observe that: ***Equipping agentic search-augmented reasoning with graph-aware designs substantially benefits graph tasks***. While Search-o1 leverages query–semantic similarity to retrieve external information, it is consistently outperformed by GraphICL, which exploits structural information through exhaustive prompt templates to identify the best-performing patterns. However, our GraphSearch method surpasses the strongest GraphICL variant on 4 out of 6 datasets, and outperforms its second-best variant on all 6 datasets. Moreover, GraphICL exhibits high sensitivity to prompt selection, requiring extensive template search that incurs significant overhead in practice, whereas GraphSearch achieves strong performance without such costly tuning.

We also conduct link prediction on three datasets, and the results in Figure 3 show that GraphSearch variants consistently outperform the baselines. On Reddit and Products, both GraphSearch-R and GraphSearch-F achieve clear gains, while GraphSearch-R delivers the best performance on Computers. These results underscore the tangible benefits of graph-aware retrieval, and we therefore

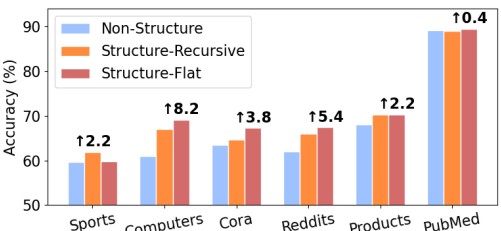

Figure 4: Node classification accuracy (%) with impact of queries w/ or w/o structure-awareness.

Table 2: Node classification accuracy (%) for *GraphSearch-R* and *GraphSearch-F* under different $\alpha$ values, with variance across $\alpha$.

| $\alpha$ | Products | Sports | Computers | Reddit | PubMed | Cora |
|---|---|---|---|---|---|---|
| *GraphSearch-R* | | | | | | |
| 0 | 70.2 | **61.9** | 67.0 | 66.0 | 89.0 | 64.6 |
| 0.5 | 69.1 | 59.8 | 68.0 | 63.5 | 89.7 | 64.8 |
| 1 | **70.8** | 60.6 | **68.1** | **66.1** | **90.4** | **65.9** |
| *GraphSearch-F* | | | | | | |
| 0 | 70.3 | **59.8** | 69.1 | **67.4** | 89.5 | 67.3 |
| 0.5 | **71.7** | 58.8 | 68.9 | 65.1 | **89.8** | 65.7 |
| 1 | 70.5 | 58.1 | **69.9** | 65.0 | 89.7 | **67.5** |
| Variance (R) | 0.50 | 0.75 | 0.25 | 1.45 | 0.33 | 0.29 |
| Variance (F) | 0.38 | 0.49 | 0.19 | 1.23 | 0.02 | 0.65 |

argue that GraphSearch is well-suited for real-world graph scenarios, combining effectiveness with minimal reliance on prompt engineering.

**Comparison with More Graph Learning Methods and GraphRAGs.** To assess the potential of agentic, search-augmented reasoning in zero-shot graph learning, we compare *GraphSearch-R* against standard graph learning methods under both zero-shot and supervised settings. As shown in Appendix A.2.2, *GraphSearch consistently outperforms traditional graph learning methods by over +50% accuracy on all three target graphs*, and *training-free methods like GraphICL, Search-o1, and GraphSearch perform comparably to supervised GNN/GraphLLM approaches*.

We also apply GraphRAG-style methods to show their performance in graph learning. As shown in Appendix A.2.4, *even strong GraphRAG with GPT4 series models as the backbone underperform by GraphSearch with Qwen2.5-72B-Instruct*.

### 4.3 OBLIGATION STUDY (RQ2)

**Effectiveness of Structure Awareness in Queries.** We fix ranking to semantic-only ($\alpha=0$) and vary only the query templates with or without structural awareness. Structure-aware queries are able to encode search scope (e.g., hop constraints) and guide neighbor–candidate construction by the retriever. As shown in Fig. 4, they outperform non-structural queries across all datasets, specifically in the Computers dataset by **+ 8.2**(%). This validates that disentangling graph structural signals from natural language semantics in queries yields clear benefits for graph tasks.

**Impact of Ranking Strategies.** We examine how varying $\alpha$ (Sec. 3.3.2) affects performance across datasets and traversal modes (Table 2). GraphSearch-R consistently performs best at $\alpha = 1$ (pure anchor node attribute), as recursive traversal already captures structure, making anchor semantics more decisive. GraphSearch-F benefits from tuning $\alpha$, reflecting its need to balance structure and semantics. Sensitivity to $\alpha$ also aligns with graph density: high-degree graphs (e.g., Reddit, Sports) show greater variance, while sparse graphs (e.g., PubMed, Cora, Products) are less affected due to smaller neighborhoods.

**Impact of Various Backbones.** We evaluate Graph-Search with diverse backbones, including a smaller 7B model, a LLaMA-series model, and a stronger open-source model. Results and analysis in Appendix A.2.3 validates consistent gains, confirming the robustness of our framework across different backbones.

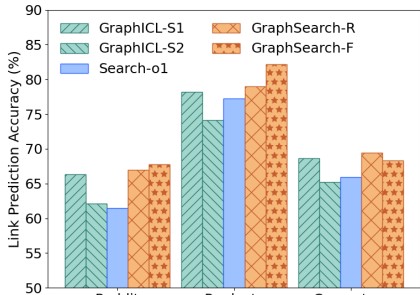

Figure 3: Link prediction accuracy (%) with the zero-shot models of GraphSearch and baselines on three datasets. Qwen2.5-32B-Instruct is the backbone.

### 4.4 LEARNABLE EXTENSION OF GRAPHSEARCH (RQ3)

**Experiment Setup.** To address **RQ3**, we explore a fine-tuned variant using lightweight reinforcement learning (RL) to jointly optimize the planning and retrieval decisions of the LRM. Specifically,

we adopt GraphSearch-F with Qwen2.5-7B-Instruct as the backbone and train it via Generalized Policy Optimization (GRPO) with a policy-gradient objective. The reward integrates (1) reasoning format consistency, (2) final prediction accuracy, and (3) penalties for omitting search. Implementation details and training configurations are provided in Appendix A.2.4.

**Results and Analysis.** Figure 5 demonstrates the effectiveness of adapting GraphSearch to supervised training settings. The RL-tuned GraphSearch variant improves performance across all datasets, achieving gains from **+2.7%** to **+10.7%**. The largest improvements are on Products (**+10.4%**) and Reddit (**+10.7%**), where agentic search plays a critical role. Even with a simple reward design, notable improvements are observed. These results highlight that *GraphSearch can be effectively extended to learnable settings, yielding consistent and substantial performance improvements.*

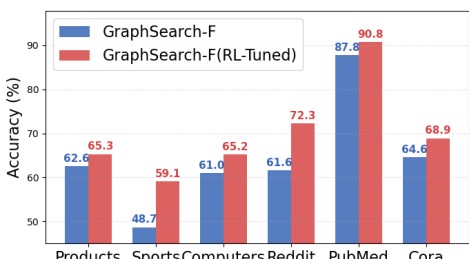

Figure 5: Comparison of GraphSearch variants with training-free and GRPO training versions.

### 4.5 EFFICIENCY ANALYSIS (RQ4)

**Retriever Efficiency Analysis.** We evaluate efficiency using the *average per-retrieval time* (mean over 5,000 test cases), comparing our graph-aware retriever GraphSearch with a non-structural baseline (Search-o1). As shown in Fig. 6, GRAPHSEARCH consistently reduces retrieval latency across all six datasets, achieving **1.29–5.77×** speedups with a geometric-mean speedup of **3.06×**. These improvements demonstrate that structure-aware candidate construction and ranking substantially lower search cost by narrowing the search space beyond query semantics alone. Our current prototype employs a simple in-memory dictionary index, and we expect further gains with more optimized indexing strategies.

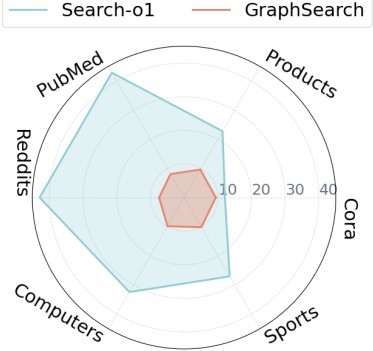

Figure 6: Time per retrieval process.

**Token Usage with Break Down.** To further assess efficiency, in Appendix A.2.4, we compare the total token usage of GraphSearch with the strong baseline Search-o1, decomposed into four reasoning phases: *think, search, information, and answer*. Results show that *GraphSearch maintains comparable token usage across all stages without incurring additional overhead.*

## 5 CONCLUSION

In this work, we introduced GraphSearch, the first framework that extends agentic search-augmented reasoning in graph learning. Unlike existing approaches that treat queries in isolation or rely on static neighbor injection, GraphSearch dynamically incorporates structural priors into reasoning by combining a Graph-Aware Query Planner with a Graph-Aware Retriever. This disentangled design enables LRMs to issue graph-expressive queries, construct topology-grounded candidate sets, and robustly rank neighbors through hybrid scoring. We further instantiated GraphSearch in two complementary variants: GraphSearch-R, which performs recursive hop-by-hop exploration, and GraphSearch-F, which supports flexible local-to-global retrieval without hop constraints. Extensive experiments on diverse benchmarks for node classification and link prediction demonstrated that GraphSearch achieves super-competitive results against state-of-the-art graph learning and search-augmented reasoning baselines, while delivering notable gains in efficiency and robustness. By bridging semantic reasoning with structural awareness, GraphSearch opens a new direction for zero-shot graph learning and paves the way toward more generalizable, efficient, and trustworthy reasoning systems over structured knowledge.

## 6 ETHICS STATEMENT

There are no ethical issues of this study.

## 7 REPRODUCIBILITY STATEMENT

Our work can be easily reproduced. We adopt multiple open-source datasets, with their descriptions provided in Appendix A.2.1. We also present detailed search strategies in Sec 3 and prompt templates in Appendix A.3. Once the paper is accepted, we will release our code.

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

## A APPENDIX

### A.1 LLM USAGE

In this paper, LLM is used for polishing to avoid grammar mistakes and make the expression smoother.

### A.2 EXPERIMENT DETAILS

In Table 1, GraphSearch and baselines are set with the temperature to be 0.7 for all backbone models and the max token length 8192. In our implementation, if the planner emits an invalid structural scope (e.g., undefined mode or out-of-range hop), we default to 'local' mode with hop=1. The retriever selects the top 3 highest-scoring node information from the graph data. We consider local neighbors up to 2 hops. All experiments are conducted on 2 NVIDIA A100-80G GPUs.

Table 3: Datasets Statistics

| Datasets | #Nodes | #Edges | #Avg.Degree | # Classes | Domain |
|---|---|---|---|---|---|
| Products (subset) | 54,025 | 74,420 | 2.75 | 47 | E-commerce |
| Sports | 173,055 | 1,946,555 | 22.48 | 13 | E-commerce |
| Computers | 87,229 | 808,310 | 18.54 | 10 | E-commerce |
| Reddit | 13,037 | 566,160 | 86.91 | 20 | Social Network |
| PubMed | 19,717 | 44,338 | 4.50 | 3 | Citation Network |
| Cora | 2,708 | 5,429 | 4.01 | 7 | Citation Network |

### A.2.1 DATASET STATISTICS

Table 3 shows the datasets statistics with detailed graph structure and domain categories.

### A.2.2 Comparison of Zero-Shot and Supervised Methods

We set GraphSearch with Qwen2.5-32B-Instruct backbone and compare it with traditional graph learning methods, following a standardized transfer protocol in which they are trained or fine-tuned on a source graph (Cora or PubMed) and then evaluated on a disjoint target graph (Products, Sports, or Computers) with no node or edge overlap. The results in Table 4 demonstrate that GraphSearch, through its agentic LRM-based framework, significantly outperforms traditional GNNs in zero-shot settings, consistent with well-known findings that GNN models exhibit poor transferability across graphs.

Table 4: Cross-Domain results of node classification. (%)

| Model | Products | Sports | Computers |
|---|---|---|---|
| GCN | 2.89 | 14.78 | 16.56 |
| SAGE | 1.11 | 5.56 | 3.89 |
| GraphPrompter | 15.42 | 9.26 | 26.40 |
| GraphSearch-R | 70.80 | 61.90 | 68.10 |

Table 5: Semi-supervised classification results of standard graph neural networks and LLM-based graph language models. These results are copied from Sun et al. (2025b). **Bold** values indicate the best performance while underlines denote the second best.

| Model | Products | Sports | Computers | PubMed | Cora |
|---|---|---|---|---|---|
| MLP | 65.36 | 58.74 | 44.56 | 59.38 | 47.23 |
| GCN | 74.47 | 70.24 | 59.12 | 74.25 | **68.82** |
| SAGE | 72.35 | 69.53 | 58.52 | 64.66 | 64.58 |
| RevGAT | 71.45 | 64.63 | 55.48 | 64.10 | 65.31 |
| LLaGA-ND | 73.32 | 62.19 | 49.48 | 39.96 | 48.52 |
| LLaGA-HO | 72.76 | 63.81 | 55.68 | 40.37 | 40.96 |
| GraphPrompter | **76.34** | **80.92** | 62.46 | 88.11 | 51.11 |
| GraphTranslator | 41.32 | 22.88 | 38.95 | 60.46 | 35.59 |
| GraphSearch-R-32B | 70.80 | 61.90 | 68.10 | **90.40** | 65.90 |
| GraphSearch-F-32B | 71.70 | 59.80 | _69.90_ | 89.80 | 67.50 |
| GraphSearch-R-72B | _74.60_ | _72.00_ | 69.60 | 89.90 | _68.80_ |
| GraphSearch-F-72B | 73.20 | 69.20 | **71.00** | _90.00_ | 67.20 |

### A.2.3 GraphSearch with Different Backbones

As shown in Table 6, GraphSearch yields consistent gains over Search-o1 across both Qwen2.5-7B-Instruct, Llama-3.3-70B-Instruct, and DeepSeek-R1 (by calling API: https://api.deepseek.com/v1). With the smaller 7B backbone, GraphSearch-R and GraphSearch-F already surpass Search-o1 on most datasets, indicating that effective structural reasoning can be achieved even with limited LRM capacity. When equipped with a stronger backbone, such as DeepSeek-R1, GraphSearch further amplifies these improvements and achieves the best results across nearly all tasks. These findings show that GraphSearch functions as a model-agnostic agentic procedure whose benefits generalize across LRM scales rather than relying on high-capability models.

### A.2.4 GraphSearch with RL-Training

In Table 7, we show the GRPO training hyperparameters used in our experiments.

Table 6: GraphSearch with different backbone LRMs.

| Model | Products | Sports | Computers | Reddit | PubMed | Cora |
|---|---|---|---|---|---|---|
| **Qwen2.5-7B-Instruct** | | | | | | |
| Search-o1 | 65.3 | 47.8 | 57.2 | 55.6 | 89.1 | 57.9 |
| GraphSearch-R | 65.2 | **53.3** | 60.4 | 60.6 | **89.6** | 59.8 |
| GraphSearch-F | 62.6 | 48.7 | **61.0** | **61.6** | 89.3 | **64.6** |
| **Llama-3.3-70B-Instruct** | | | | | | |
| Search-o1 | 75.5 | 73.6 | 66.8 | 66.3 | 90.9 | 65.5 |
| GraphSearch-R | 76.9 | 76.1 | 71.9 | 69.8 | 90.6 | 69.8 |
| GraphSearch-F | 78.7 | 75.3 | 70.3 | 71.1 | 91.0 | 69.7 |
| **DeepSeek-R1** | | | | | | |
| Search-o1 | 73.0 | 79.3 | 64.9 | 74.7 | **90.3** | 69.0 |
| GraphSearch-R | 74.3 | 80.1 | 74.9 | 80.7 | 89.8 | **72.1** |
| GraphSearch-F | **75.6** | **80.2** | **75.1** | **81.7** | 89.5 | 72.0 |

Table 7: RL training configuration for GraphSearch-F.

| Hyperparameter | Value |
|---|---|
| Backbone LRM | Qwen2.5-7B-Instruct |
| RL algorithm | Policy Gradient |
| Reward components | Format + Accuracy + Search Usage Penalty |
| Learning rate | 1e-5 |
| Batch size | 4 |
| Training steps | 2,000 |

### A.2.5 GRAPHSEARCH COMPARISON TO GRAPHRAG METHODS

**Implementation Details.** For a fair comparison, we implemented GraphRAG and HippoRAG2 under a unified protocol. We adopt a chunk size of 1200 tokens with 100-token overlap for all text documents. For HippoRAG2, we use `NV-embed-v2` as the embedding model and `GPT-4o-mini` as the LLM backbone. For GraphRAG, we use `text-embedding-3-small` as the embedding model and `GPT-4-Turbo-Preview` as the backbone. All methods retrieve the top-5 documents per query and follow our zero-shot evaluation protocol.

**Analysis.** Across both datasets, GraphRAG-style methods underperform GraphSearch by a clear margin. This is expected because GraphRAG constructs a chunk-level co-occurrence graph and does not utilize the native graph topology, which is crucial for node- and edge-level prediction. Even with strong GPT-4-series backbones, GraphRAG and HippoRAG2 fail to transfer effectively to structure-dependent graph learning tasks, while GraphSearch, operating directly on the original graph structure, achieves consistently superior accuracy.

Table 8: GraphSearch comparison to GraphRAG and its variants. (%)

| Model | Reddit | Cora |
|---|---|---|
| GraphRAGE (GPT4-Turbo) | 62.4 | 61.4 |
| HippoRAG2 (GPT4-Turbo) | 65.0 | 64.8 |
| GraphSearch-R (Qwen2.5-72B) | 66.6 | 68.8 |
| GraphSearch-F (Qwen2.5-72B) | 68.7 | 67.2 |

### A.2.6 MORE GRAPHSEARCH EFFICIENCY ANALYSIS

Based on over 7,000 data points, we provide a detailed breakdown of token usage across reasoning phases, directly comparing GraphSearch-R with Search-o1 under the identical model backbone, Qwen2.5-32B-Instruct.

**Total Token Usage Across Datasets.** Table 9 summarizes the average per-example token usage on six datasets.

Table 9: Average token count (per example) on six datasets.

| Model | Products | Sports | Computers | Reddit | PubMed | Cora |
|-------|----------|--------|-----------|--------|--------|------|
| Search-o1 | 2020 | 1364 | 2002 | 1055 | 2567 | 2524 |
| GraphSearch-R | 2039 | 1336 | 1967 | 1229 | 2643 | 2764 |

**Phase-level Token Breakdown.** We further break down token usage by reasoning phase and report the average proportion across all six datasets. Results are shown in Table 10.

Table 10: Token distribution across reasoning phases (averaged over six datasets).

| Model | think | search | information | answer |
|-------|-------|--------|-------------|--------|
| Search-o1 | 45.03% | 4.25% | 49.86% | 0.86% |
| GraphSearch | 46.17% | 4.03% | 48.96% | 0.67% |

**Discussion.** Across all datasets, GraphSearch-R and Search-o1 show **very similar total token lengths and nearly identical phase-level distributions** (search ≈4%, answer < 1%). This indicates that our graph-aware planner and retriever do not introduce additional reasoning overhead, since they change *which* nodes are retrieved, not *how much* content is retrieved.

### A.3 PROMPT TEMPLATE FOR GRAPHSEARCH

Table 11: Prompt template used for GraphSearch-F.

| Model | Prompt |
|---|---|
| GraphSearch-F | You are a reasoning assistant for node classification on an {Amazon} product graph. Your goal is to select the most likely category for the target node from the provided list.

**Tools:**
- To perform a search, use this schema exactly: `<search> mode={local|global}, hop={1|2}, query={your query with keywords} </search>`
• mode=local: recall neighbors within 1–2 hops of the target node and you must specify hop=1 or hop=2
• mode=global: recall from a global nodes pool (e.g., PageRank)
- The graph retriever considers both the graph structure and the semantic similarity of your query, and returns the most relevant data inside `<information> ... </information>`.

**Reasoning protocol:**
- Begin with `<think>...</think>` to assess if attributes and graph stats are sufficient.
- Whenever you receive new information, first reason inside `<think> ... </think>`.
- If no further information is needed, output only your final choice inside `<answer> ... </answer>` (no extra explanation).

**Decision Policy:**
- Rich attributes → predict after one search.
- Weak/incomplete → mode=local, hop=1; if ambiguous and degree moderate/high → mode=local, hop=2.
- Very low degree or conflicting neighbors → mode=global.

**Example:**
Question: ...
Assistant:
`<think> ...your reasoning...</think>`
`<search> ...your query... </search>`
`<information>...retriever results...</information>`
`<think>...reasoning with the new information...</think>`
`<answer>Movies</answer>`

**Use the following information for the node classification task:**
- The target [product]'s information: {title and description}
- The domain knowledge: Each node represents a product and connected to other products through co-purchase relationships. The degree of target node is {}, while the average degree of the dataset is {}.
- The category list: {...; ...; ...;} |

Table 12: Prompt template used for GraphSearch-R.

| Model | Prompt |
|---|---|
| GraphSearch-R | You are a reasoning assistant for node classification on an {Amazon} product graph. Your goal is to select the most likely category for the target node from the provided list.

**Tools:**
- To perform a search, write `<search> your query here </search>`.
- The graph retriever considers both the graph structure and the semantic similarity of your query, and returns the most relevant data inside `<information> ... </information>`.
- You can repeat the search process multiple times if necessary.

**Reasoning protocol:**
- Whenever you receive new information, first reason inside `<think> ... </think>`.
- If no further information is needed, output only your final choice inside `<answer> ... </answer>` (no extra explanation).

**Example:**
Question: "..."
Assistant:
`<think> ...your reasoning...</think>`
`<search> ...your query... </search>`
`<information>...retriever results...</information>`
`<think>...reasoning with the new information...</think>`
`<answer>Movies</answer>`

**Use the following information for the node classification task:**
- The target [product]'s information: {title and description}
- The domain knowledge: Each node represents a product and connected to other products through co-purchase relationships. The degree of target node is {}, while the average degree of the dataset is {}.
- The category list: {...; ...; ...;} |