# OpenReview forum: "GraphSearch: Agentic Search-Augmented Reasoning for Zero-Shot Graph Learning"
_ICLR.cc/2026/Conference — Submitted to ICLR 2026_

### Official Review · Reviewer_VWwh · 2025-10-22

**Soundness:** 3
**Presentation:** 2
**Contribution:** 2
**Rating:** 4
**Confidence:** 3

**Summary:**

The paper proposes GraphSearch, an agentic framework that equips an LRM with a graph-aware query planner that disentangles where to search from what to search for, and a Graph-aware Retriever that constructs structurally grounded candidate sets and ranks them with a hybrid score combining anchor–attribute similarity and query–semantic similarity. Experiments on six datasets for node classification (and three for link prediction) show competitive zero-shot performance vs. Search-o1, GraphICL, and supervised GNN/GraphLLM references, with reported efficiency gains in retrieval latency.

**Strengths:**

- Identification and separation between structure specification and semantic intent in queries is an interesting observation and the handling with a separate module is well-crafted
- Ablations on structure-aware queries vs. non-structural prompts and α-sensitivity are helpful
- Efficiency analysis is a nice touch

**Weaknesses:**

- The paper over-claims novelty as “the first framework” for agentic search on graphs; prior work, such as Graph-CoT[1] and RoG[2], already integrates planning, retrieval, and reasoning over graph structures. Please discuss and compare with the existing baseline empirically and how they differ from the proposed method conceptually.

- Limited baselines and base models: comparisons omit other agentic retrieval augmentation frameworks (e.g., GraphRAG, G-Retriever, ReAct) and stronger LLMs such as Llama-3.1, Qwen-2.5, or Gemma-2, making it unclear how robust the proposed approach truly is and how generalizable it is.

- The retrieval scoring relies largely on cosine similarity of attributes, raising concerns that structural information only constrains the candidate set rather than contributing meaningfully to ranking.


- No public code or detailed configuration is provided, hindering reproducibility.



[1] Jin, Bowen, et al. "Graph chain-of-thought: Augmenting large language models by reasoning on graphs." arXiv preprint arXiv:2404.07103 (2024).
[2] Luo, Linhao, et al. "Reasoning on graphs: Faithful and interpretable large language model reasoning." arXiv preprint arXiv:2310.01061 (2023).

**Questions:**

- How exactly is structural information integrated into the final ranking beyond attribute cosine similarity?
- Could the planner and retriever be jointly optimized or adapted via reinforcement learning or feedback, similar to RLVR or Search-o1?

---

> ### Author Response · Authors · 2025-12-03
>
> **W1**: *Novelty claim and more existing agentic method comparison.*
>
> **R1**: We appreciate the reviewer’s observation and agree that our previous phrasing may not have precisely captured the scope of our contribution. Our intention is not to claim the first agentic method in general, but to highlight a more specific novelty: **GraphSearch is the first framework that extends search-augmented reasoning into graph learning through graph-guided, multi-step agentic rollouts.**
>
> **Graph-CoT is designed for factoid QA with structured knowledge sources rather than graph learning tasks.** An LLM answers natural-language questions by issuing predefined function calls to retrieve specific nodes, their features, or their neighbors. Its planning and retrieval rely on fixed API calls without any filtering mechanism, meaning there is no graph-aware query generation or top-k selection. As a result, Graph-CoT often returns full neighbor sets and the LLM reasons over all of them, which leads to longer inference chains and significantly higher latency. In contrast, GraphSearch formulates graph-aware queries, applies ranked top-k structural retrieval, and uses the retrieved subsets for zero-shot node and edge prediction.
>
> **Empirically, we evaluate GraphCoT using Qwen2.5-32B-Intsrtuc as the backbone and compare it with GraphSearch** under the same backbone on node classification tasks (%):
>
> | Model          | Products | Sports | Computers | Reddit | PubMed | Cora |
> |----------------|----------|--------|-----------|--------|--------|-------|
> | Graph-CoT      | 64.6     | 57.2   | 62.6      | 56.1   | 70.5   | 57.9  |
> | GraphSearch-R  | 70.8     | **61.9**   | 68.1      | 63.5   | **90.4**   | 65.9  |
> | GraphSearch-F  | **71.7**     | 59.8   | **69.9**      | **67.4**   | 89.8   | **67.5**  |
>
> **GraphSearch outperforms Graph-CoT in all datasets** by +4.7 to +19.9 percent points, with the largest gains on PubMed(+19.9) and Reddit(+11.3). We also observe that GraphCoT’s unfiltered neighbor retrieval incurs substantial overhead, resulting in roughly **3× slower inference** compared with GraphSearch.
>
> **RoG performs one-step relation-path planning for KG-QA queries, which cannot transfer to standard graph learning tasks.** Node classification provides only a single node, leaving no relation path to plan, and link prediction may involve nodes absent from the graph, making paths impossible to ground. Because these tasks fundamentally differ from QA, RoG’s planning mechanism cannot extend to zero-shot node or edge prediction.
>
> We have clarified our positioning statement in *Introduction*, revised the *Related Work*, and added evaluation results in *Table 1*.

---

> > ### Author Response · Authors · 2025-12-03
> >
> > **W2**: *Limited GraphRAG baselines and base models*
> >
> > **R2**: *(i) Comparison of GraphRAG and its variant.*
> >
> > **Our task setting fundamentally differs from GraphRAG-style QA systems.** We focus on graph-guided agentic reasoning for prediction-driven graph learning tasks (e.g., node classification, link prediction), where leveraging the original graph structure is essential. In contrast, GraphRAG and its variants operate over constructed graphs formed by chunk-level entity co-occurrence, and do not use native graph topology. Our method reasons directly over existing graphs, injecting structural context into both retrieval and planning.
> >
> > **For completeness, we implemented GraphRAG and HippoRAG2 with GPT4 series models as the backbone under our evaluation protocol**. As shown below, these models consistently underperform in standard graph learning settings on node classification tasks (%):
> >
> > | Model                     | Reddit | Cora  |
> > |---------------------------|--------|-------|
> > | GraphRAG (GPT4o-mini)     | 62.36  | 61.42 |
> > | HippoRAG2 (GPT4-Turbo)    | 65.00  | 64.84 |
> > | GraphSearch-R (Qwen2.5-72B) | **66.6** | **68.8** |
> > | GraphSearch-F (Qwen2.5-72B) | **68.7** | **67.2** |
> >
> > Results show that **even GraphRAG variants built on GPT-4 series models perform worse than GraphSearch with Qwen2.5-72B-Instruct.**
> >
> > *(ii) GraphSearch with stronger backbones.*
> >
> > GraphSearch is a methodological framework rather than a model-specific benchmark, and it does not rely on any particular LRM. To further demonstrate that the agentic search mechanism is robust across model capacities, **we include a strong LRM Llama-3.3-70B-Instruct and a much stronger open-source model, DeepSeek-R1.**
> >
> > Backbone: Llama-3.3-70B-Instruct
> >
> > | Model         | Products | Sports | Computers | Reddit | PubMed | Cora |
> > |---------------|----------|--------|-----------|--------|--------|-------|
> > | Search-o1     | 75.5     | 73.6   | 66.8      | 66.3   | 90.9   | 65.5  |
> > | GraphSearch-F | 76.9     | **76.1**   | **71.9**      | 69.8   | 90.6   | **69.8**  |
> > | GraphSearch-R | **78.7**     | 75.3   | 70.3      | **71.1**   | **91.0**   | 69.7  |
> >
> > Backbone: DeepSeek-R1
> >
> > | Model         | Products | Sports | Computers | Reddit | PubMed | Cora |
> > |---------------|----------|--------|-----------|--------|--------|-------|
> > | Search-o1     | 73.0     | 79.3   | 64.9      | 74.7   | **90.3**   | 69.0  |
> > | GraphSearch-F | 74.3     | 80.1   | 74.9      | 80.7   | 89.8   | **72.1**  |
> > | GraphSearch-R | **75.6**     | **80.2**   | **75.1**      | **81.7**   | 89.5   | 72.0  |
> >
> > **Across all these backbone settings, GraphSearch consistently outperforms the strong baseline Search-o1**. We have included these results for robustness analysis of GraphSearch in *Appendix A.2.3*.
> >
> >
> >
> > **W3/Q1**: *How is the Structural Information Integrated into the Final Ranking?*
> >
> > **R3**: **Structural information influences both the candidate pool and the final ranking.**
> >
> > First, **the planner selects the retrieval mode** (local with specific hop size / global / attribute-only) **to determine the structural scope of the candidate set**, so the structure already constrains which nodes are eligible for ranking.
> >
> > Second, within this structurally defined neighborhood, the ranking score combines (1) the query–candidate semantic similarity and (2) **the anchor–candidate attribute similarity, which serves as a semantic affinity weight on the anchor–candidate edge**. This anchor-derived structural signal is explicitly included in the α-weighted scoring, meaning structure affects not only which nodes enter the pool but also how they are ordered.
> >
> > In short, structure guides both the formation of the candidate neighborhood and the final ranking through anchor-conditioned scoring.
> >
> >
> >
> > **W4**: *Reproducibility.*
> >
> > **R4**: We have included key implementation details in *Appendix A.2 and prompts in Appendix A.3*. We will release the full code after publication to ensure transparency.

---

> ### Author Response · Authors · 2025-12-03
>
> **Q2**: *Optimize via Reinforcement Learning.*
>
> **RQ2:**
> Our primary goal is to explore training-free agentic reasoning for graph learning, inspired by frameworks like Search-o1. This design emphasizes generalizability, enabling plug-and-play transfer across models and tasks without task-specific fine-tuning.
>
> We appreciate the suggestion and **have conducted experiments with a learnable variant trained via reinforcement learning (RL)**. Using Qwen2.5-7B-Instruct as the backbone, we apply GRPO with a reward function that combines:  (1) reasoning format consistency, (2) prediction accuracy, (3) penalties for no search. We train the model for 2,000 steps with a learning rate of 1e-5 and a batch size of 4.
>
> | Model | Products | Sports | Computers | Reddit | PubMed | Cora |
> |-------|----------|--------|-----------|--------|---------|-------|
> | GraphSearch-F | 62.6 | 48.7 | 61.0 | 61.6 | 87.8 | 64.6 |
> | GraphSearch-F (RL Trained) | **65.3** | **59.1** | **65.2** | **72.3** | **90.8** | **68.9** |
>
> **The RL-Trained model achieves consistent notable gains ranging from +2.7% to +10.7%**, validating that our agentic framework is compatible with learning-based enhancements and can further boost performance. We have updated the paper to describe this extension in *Sec 4.4* and include detailed training configurations in *Appendix A.2.4*.

---

### Official Review · Reviewer_LMYp · 2025-10-24

**Soundness:** 3
**Presentation:** 3
**Contribution:** 3
**Rating:** 6
**Confidence:** 5

**Summary:**

GraphSearch is the first framework to extend search-augmented reasoning to graphs, enabling zero-shot graph learning without task-specific fine-tuning. It has two key components and two traversal modes:
1. Graph-aware Query Planner: Separates search space (e.g., 1-hop, multi-hop neighbors) from semantic queries.
2. Graph-aware Retriever: Builds candidate sets based on graph topology and ranks them via a hybrid scoring function.
Traversal Modes:
1. GraphSearch-R: Recursively expands neighborhoods step by step.
2. GraphSearch-F: Flexibly retrieves across local/global neighborhoods without hop limits.

**Strengths:**

1. It is the first framework that extends agentic search-augmented reasoning to graph-structured data, enabling zero-shot graph learning.
2. Its core contributions include a graph-aware query planner (which decouples search space from semantic queries), a graph-aware retriever (which constructs candidate sets based on topology and uses hybrid scoring), and two traversal modes: GraphSearch-R and GraphSearch-F.
3. Extending search-augmented LRM (Large Reasoning Model) to the graph domain is a meaningful research direction, addressing the issue that existing methods cannot utilize graph structural information.
4. The design idea of decoupling search space (topology) from semantic content is clear, and the hybrid ranking function balances structural and semantic relevance.
5. Experiments on node classification and link prediction are conducted on 6 datasets, with comparisons against multiple baseline methods.

**Weaknesses:**

1. Essentially, it adapts existing search-augmented reasoning (e.g., Search-o1) to graph data, resulting in limited technical innovation.
2. It does not provide theoretical analysis on how graph structure affects reasoning.
3. It lacks a learning mechanism and fully relies on predefined prompt templates.
4. Figure 4 only demonstrates the impact of query structure awareness, with no ablation studies on other components.
5. It does not analyze the impact of different hop counts and candidate set sizes.

**Questions:**

1. Strengthen theoretical analysis and provide a formal framework for how graph structure affects reasoning.
2. Design a learning-based query planner instead of relying on fixed templates.
3. Conduct fair comparisons with the latest graph learning methods.

---

> ### Author Response · Authors · 2025-12-03
>
> **W1**: *Limited Technical Innovation*
>
> **R1**: The paradigm of search-augmented reasoning (e.g., Search-o1) is already well-established, and our contribution is not to propose a new paradigm, but, as mentioned in the reviewer strength list, to be **the first to adapt agentic, training-free search to standard graph learning tasks**. Graph learning suffers from poor transferability of supervised models, making a training-free agentic approach particularly valuable.
>
> To make search-augmented reasoning work for graph learning, we introduce **new components** that go beyond existing frameworks:
>
> - **Graph-aware planner** that dynamically selects retrieval scopes (hop-k, global, attribute), injecting structural context into the agent’s control flow rather than relying solely on semantic cues.
> - **Graph-aware retriever** that combines semantic similarity with topological alignment through a weighted strategy, enabling structure-aware ranking rather than chunk selection as in prior RAG-style methods.
> - **Two traversal modes** that mirror GNN-style recursive aggregation (GraphSearch-R) and a more flexible flat (GraphSearch-F) expansion.
>
> These components are necessary for adapting graph-aware agentic rollouts to graph learning tasks, which prior search-augmented methods do not address.
>
>
> **W2/Q1**: *Lack of Theoretical Analysis of Graph Structure*
>
> **R2**: This is a valuable and interesting point. Our study is primarily empirical, and developing theoretical characterizations is valuable future work that we plan to pursue, such as performance bounds under different graph regimes.
>
>
>
> **W3/Q2**: *Lack of Learning Mechanism*
>
> **R3**: Our primary goal is to explore training-free agentic reasoning* for graph learning, inspired by the success of prompt-based methods (e.g., Search-o1). This design enables high generalization and plug-and-play transfer across tasks and models.
>
> Following the reviewer’s suggestion, **we have conducted preliminary experiments with a learnable variant trained via reinforcement learning (RL)**. We apply GRPO and use Qwen2.5-7B-Instruct as the backbone and optimize using a reward combining:  (1) reasoning format consistency,  (2) prediction accuracy,  (3) penalties for no search. We train the model for 2,000 steps with a learning rate of 1e-5 and a batch size of 4.
>
> | Model                   | Products | Sports | Computers | Reddit | PubMed | Cora |
> |-------------------------|----------|--------|-----------|--------|--------|-------|
> | GraphSearch-F           | 62.6     | 48.7   | 61.0      | 61.6   | 87.8   | 64.6  |
> | GraphSearch-F (RL-Trained) | **65.3** | **59.1** | **65.2** | **72.3** | **90.8** | **68.9** |
>
> **The RL-Trained model achieves consistent notable gains ranging from +2.7% to +10.7%**, validating that our agentic framework is compatible with learning-based enhancements.
>
> We have updated the paper to describe this extension and include implementation details, results, and analysis in *Sec 4.4* and *Appendix A.2.4*.
>
> **W4**: *Missing Ablations on Other Components*
>
> **R4**: In **Table 2** of the paper, we vary the weighting between structural and semantic similarity in our hybrid ranking module to study the retriever’s ranking behavior. These results demonstrate how different combinations of structure vs. semantic alignment affect retrieval quality and downstream performance, thereby highlighting the influence of the retriever’s **ranking mechanism**.
>
> **W5**: *Impact of Hop Counts & Candidate Set Sizes*
>
> **R5**: **In GraphSearch-R, hop count is not a tunable hyperparameter in our recursive setting**, since each retrieval step naturally expands the neighborhood based on the planner’s decision.
>
> **In GraphSearch-F, increasing the maximum allowable hop has a negligible effect**. We expanded the local mode to allow hop ∈ {1, 2, 3} and evaluated the planner’s behavior on over 7,000 test cases. The planner selected hop=3 **only once**, overwhelmingly choosing between hop=1 and hop=2. This suggests that larger hops offer little additional utility.
>
> This behavior is consistent with established graph learning practice, where 'local' neighborhoods almost always refer to 1–2 hop ranges. For this reason, our 'mode=local' setting uses hop 1 or 2.

---

> ### Author Response · Authors · 2025-12-03
>
> **Q3**: *Comparisons with the Latest Graph Learning Methods*
>
> **R6**:  **We further expand the baseline coverage by including both GNN-based methods (GCN, GraphSAGE) and LLM-based prompting methods (GraphPrompter)**, ensuring that our comparisons span the two major families of graph learning approaches.  To ensure fair zero-shot evaluation, we adopt a standardized protocol where all models are trained or fine-tuned on one graph set (Cora/PubMed) and evaluated on a disjoint target set (Products/Sports/Computers), with no overlap. This follows the established zero-shot setting used in prior work, such as GraphICL, which is already included in our paper.
>
> | Model         | Products | Sports | Computers |
> |---------------|----------|--------|-----------|
> | GCN           | 2.89     | 14.78  | 16.56     |
> | GraphSAGE     | 1.11     | 5.56   | 3.89      |
> | GraphPrompter | 15.42    | 9.26   | 26.40     |
> | GraphSearch-R | **70.8** | **61.9** | **68.1** |
>
> These results demonstrate that **GraphSearch, leveraging an agentic framework with LLMs, significantly outperforms (with +50% improvements) traditional GNN-based and LLM-based baselines in zero-shot scenarios**.
>
> We have included these results and analyses in *Sec 4.2* and *Appendix A.2.2*.

---

### Official Review · Reviewer_yBUY · 2025-10-31

**Soundness:** 3
**Presentation:** 3
**Contribution:** 3
**Rating:** 4
**Confidence:** 4

**Summary:**

The paper introduces GraphSearch, a framework that extends search-augmented large reasoning models (LRMs) to graph-structured data for zero-shot node classification and link prediction. The system separates where to search (graph topology) from what to search (semantic intent) via a Graph-aware Query Planner and a Graph-aware Retriever. Two traversal modes are instantiated: GraphSearch-R (recursive, hop-by-hop expansion) and GraphSearch-F (flexible retrieval over local/global neighborhoods without hop constraints). Experiments on six datasets suggest GraphSearch reaches competitive or better results than supervised graph learners and achieves state-of-the-art in zero-shot settings.

**Strengths:**

1.Bridging search-augmented reasoning with graphs is important for domains (e-commerce, social, citations) where topology carries critical priors that plain-text RAG overlooks.

2.Disentangling topological scope from semantic query is a clean design that can reduce retrieval noise and focus computation on structurally relevant regions.

3.Demonstrating competitive zero-shot graph learning results on node classification and link prediction, is noteworthy and of practical interest to agents that must operate on new graphs without retraining.

**Weaknesses:**

1.Much of the method reads as policy design (planner prompts, scope flags, hybrid scoring) rather than a fundamentally new retrieval or reasoning mechanism. The technical contribution should be highlighted.

2.The compared methods are limited given the claim of “first framework” and SOTA zero-shot results. Include: (i) planner–executor RAG on graphs/text, (ii) dense–sparse hybrid retrievers with structural priors, (iii) recent GraphRAG variants, and (iv) supervised GNNs tuned under equal budget. Report identical token/latency budgets and prompt templates for fairness.

3.It would be better to analyze failure modes when the planner emits incorrect structural scopes or when graphs are noisy/sparse.

4.The performance of GraphSearch-R and GraphSearch-F varies significantly across different datasets, and there is a lack of in-depth analysis and explanation for this phenomenon.

5.Zero-shot claim needs a stronger evaluation. Current tasks and datasets are relatively standard. How about compositional, long-range, and heterophilic graphs where structural cues can conflict with semantics?

6.The efficiency evidence is limited. It would be better to break down token usage and latency across think, search, information, answer phases, and compare against chunk-based search with compressed contexts.

7.Minor Typos:
Lines 140–142: repeated sentences—please remove.
Table 4: the Best metric annotation is incorrect on the Products dataset.

**Questions:**

How about the performance of GraphSearch with smaller open-source LLMs (e.g., Qwen2.5-7B)?

---

> ### Author Response · Authors · 2025-12-03
>
> **W1**: *Novelty Beyond Policy Design?*
>
> **R1**: The general paradigm of search-augmented reasoning (e.g., Search-o1) is already well-known, and our goal is to **adapt this agentic reasoning mechanism to graph learning**, which is brand-new. To make this possible, we introduce components that do not exist in prior agentic-search systems:
>
> - **Graph-aware planner** that determines structural retrieval scopes (hop-k, global, attribute). This changes the search space itself, allowing the agent to reason over graph neighborhoods rather than only over textual chunks.
> - **Graph-aware hybrid retriever** that integrates semantic relevance with topological alignment, enabling structure-aware ranking, not just chunk-level filtering as in prior RAG pipelines.
> - **Two traversal modes** that enable agentic rollouts to follow either recursive GNN-style aggregation or a flexible flat expansion, providing reasoning paths uniquely tailored to graph learning tasks.
>
> These components go **beyond prompt or policy design**: they establish a **new structural search mechanism** that prior search-augmented methods cannot support. This mechanism is essential for adapting agentic reasoning to node and edge prediction tasks that lie outside the scope of existing frameworks.
>
>
>
>
> **W2**: *More Baselines*
>
> **R2**: We appreciate the reviewer’s thoughtful suggestions. Below, we clarify how our current experiments already address these concerns and extend more baselines, including the RAG methods and GNN ones.
>
> **(i) Our evaluation (Table 1 in the paper) has included both one-shot and multi-round planner–executor RAG paradigms.** Specifically, GraphICL represents a one-time static RAG, while Search-O1 follows a multi-step agentic planning strategy. These baselines span the range of RAG depth and control structure across planning and execution, aligning with the intended contrast in this category.
>
> **(ii) Our retriever itself has covered dense-sparse hybrid variants (Table 2 in the paper)**: the retriever operates on a sparse, structure-defined candidate set (e.g., hop-k neighbors or PPR-expanded nodes), and then applies dense semantic ranking over this set, based on the LRM-generated query and the anchor node’s attributes. Our α-ablation varies the contribution of anchor-based vs. query-based similarity, showing that retrieval performance changes systematically with different mixtures of semantic signals.
>
> **(iii) We include GraphRAG variants for comparison**. Note that our task setting fundamentally differs from GraphRAG-style QA systems. We focus on graph-guided agentic reasoning for prediction-driven graph learning tasks (e.g., node classification, link prediction), where leveraging the original graph structure is essential. In contrast, GraphRAG and its variants operate over constructed graphs formed by chunk-level entity co-occurrence, and do not use native graph topology.
>
> For completeness, we implemented GraphRAG and HippoRAG2 with GPT4 series models as the backbone under our evaluation protocol. As shown below, these models consistently underperform in standard graph learning settings on node classification tasks (%):
>
> | Model             | Reddit | Cora |
> |---------------|--------|-------|
> | GraphRAG (GPT4o-mini)     | 62.36  | 61.42 |
> | HippoRAG2 (GPT4-Turbo)    | 65.00  | 64.84 |
> | GraphSearch-R (Qwen2.5-72B) | **66.6**   | **68.8**  |
> | GraphSearch-F (Qwen2.5-72B) | **68.7**   | **67.2**  |
>
> From the results, we see that **even strong GraphRAG and HippoRAG2 with GPT4 series models as the backbone are underperformed by GraphSearch with Qwen2.5-72B-Instruct.**
>
> We ensure fairness across all retrieval-based methods by controlling the maximum input token per round, while exact prompt templates differ due to structural paradigm differences (e.g., static vs. agentic). We have included the implementation details, results, and analysis in *Appendix A.2.5*.
>
> **(iv) We also include GNNs for comparison**, evaluated under the same zero-label budget. We include classical GNN models (GCN and GraphSAGE) and follow a standardized transfer protocol in which they are trained or fine-tuned on a source graph (Cora or PubMed) and then evaluated on a disjoint target graph (Products, Sports, or Computers) with no node or edge overlap.
>
> | Model       | Products | Sports | Computers |
> |-------------|----------|--------|-----------|
> | GCN         | 2.89     | 14.78  | 16.56     |
> | GraphSAGE   | 1.11     | 5.56   | 3.89      |
> | GraphSearch-R | **70.8**     | **61.9**   | **68.1**      |
>
> These results demonstrate that **GraphSearch, through its agentic LRM-based framework, significantly outperforms (more than +50% improvement) traditional GNNs in zero-shot settings**, consistent with well-known findings that GNN models exhibit poor transferability across graphs. We have included these results in *Appendix A.2.2*.

---

> > ### Author Response · Authors · 2025-12-03
> >
> > **W3**: *Failure Modes Under Incorrect Scopes or Noisy Graphs*
> >
> > **R3**: When the planner emits incorrect structural scopes, **our implementation safely defaults to 'local' mode with hop=1**. Across over 8,000 test cases, such fallbacks are triggered in only *0.3%* (Qwen2.5-32B-Instruct) and *0.1%*(Qwen2.5-72B-Instruct), indicating strong planner reliability across different backbones.
> >
> > In **sparse or noisy graphs**, the planner adapts by switching to modes like **'global'** or **'attribute'**. For example, in cases with low node degree, it emits a global query to retrieve a broader context. Here is a toy example from the testing data:
> >
> > `<think>Considering the low node degree, a global search may help to capture broader relationships and provide a more informed classification...</think><search> mode=global, query="Winchester Adjustable Closure Americana Mesh Back Cap, Realtree Xtra Camo" </search>`
> >
> > We have updated the default behaviors in the revised *Appendix A.2*.
> >
> >
> >
> >
> > **W4**: *Variance Between GraphSearch-R and GraphSearch-F*
> >
> > **R4**: Overall, **the performance differences between GraphSearch-R and GraphSearch-F are moderate and well within a reasonable range**. As shown in the table below, the average gap across two model scales remains within 1–2 percentage points for 5 out of 6 datasets, which we believe does not substantiate the claim of “significant variance.”
> >
> > | Model                           | Products | Sports | Computers | Reddit | PubMed | Cora |
> > |---------------------------------|----------|--------|-----------|--------|--------|-------|
> > | Avg Accuracy Difference (R vs. F) | 1.15     | 1.85   | 1.6       | 3.0    | 0.35   | 1.6   |
> > | Avg Node Degree                  | 2.75     | 22.48  | 18.54     | 86.91  | 4.5    | 4.01  |
> >
> > The only notable exception is Reddit, where a larger gap (~3.0) is observed. We attribute this to the graph’s structural properties: **Reddit has an exceptionally high average node degree (86.91), which magnifies differences between retrieval strategies.**
> >
> > GraphSearch-R follows a GNN-style recursive aggregation based on local neighborhoods, while GraphSearch-F enables flatter, more global exploration. In dense graphs, especially under multi-hop settings (e.g., hop=2), these two paradigms may retrieve substantially different neighbor sets due to the abundance of candidate nodes, resulting in greater performance variation. In contrast, on sparse or moderately connected graphs, retrieval overlap is higher, and the performance gap remains minimal. Importantly,  **the variation reflects complementary retrieval philosophies rather than instability**. The two variants offer different inductive biases and together improve robustness across graph types and reasoning styles.
> >
> >
> > **W5**: *Zero-Shot Evaluation on Compositional / Long-Range / Heterophilic Graphs*
> >
> > **R5**: We follow prior LLM-based graph reasoning works (e.g., GraphICL) by using standard benchmarks where existing zero-shot baselines are defined. **The OGBN-Products dataset used in our evaluation is long-range.**
> >
> > **Compositional graphs are already covered in our setting.** In graph learning, compositionality manifests as multi-hop relational reasoning, and our framework directly supports this through structured retrieval (hop-k, PPR). Thus, GraphSearch naturally handles compositional, multi-step dependencies in long-range graphs.
> >
> > **Our method can also be applied to heterophilic graphs** because the retrieval is based on the query generated by the LRM during reasoning, and even if the structure cues conflict with semantics, in our flat GrphSearch-F version, the planner will choose the model=attribuet and generate on-demand query keywords for retrieval.

---

> ### Author Response · Authors · 2025-12-03
>
> **W6**: *Efficiency Analysis and Token Breakdown*
>
> **R6**: To align with the reviewer’s suggestion, we note that Search-o1 naturally serves as the chunk-based retrieval baseline in our setting: it retrieves node-associated text purely by semantic similarity without using any graph structure. We therefore report phase-level token usage and compare GraphSearch-R directly against Search-o1 over 5000 cases under the identical backbone.
>
> The table below summarizes the average token count on six datasets:
>
> | Model          | Products | Sports | Computers | Reddit | PubMed | Cora |
> |----------------|----------|--------|-----------|--------|--------|-------|
> | Search-o1      | 2020     | 1364   | 2002      | 1055   | 2567   | 2524  |
> | GraphSearch  | 2039     | 1336   | 1967      | 1229   | 2643   | 2764  |
>
> We also break down usage by phase, averaged over all six datasets:
>
> | Model              | think    | search  | information | answer |
> |--------------------|----------|---------|-------------|--------|
> | Search-o1          | 45.03%   | 4.25%   | 49.86%      | 0.86%  |
> | GraphSearch | 46.17%   | 4.03%   | 48.96%      | 0.67%  |
>
> Across six datasets, GraphSearch-R and Search-o1 have very similar total token lengths, and their phase-level token distribution is nearly identical. This indicates that **our graph-aware planner and retriever do not introduce additional reasoning overhead**, since they change which nodes are retrieved, not how much is retrieved.
>
> We have included these comparisons and analyses in *Appendix A.2.6*.
>
>
> **W7**: *Minor Typos and Table Corrections*
>
> **R7**: Thank you for pointing them out. We have removed the repeated sentences of Lines 140–142 and also corrected the best metric annotation in Table 4: the best is on the GraphPrompt baseline for the Product dataset.
>
>
>
> **Q1**: *Performance with Smaller Open-Source LLMs*
>
> **R8**: **We have added experiments using Qwen2.5-7B-Instruct as the backbone** to evaluate GraphSearch and the strong baseline Search-o1:
>
> | Model          | Products | Sports | Computers | Reddit | PubMed | Cora |
> |----------------|----------|--------|-----------|--------|--------|-------|
> | Search-o1      | 63.3     | 47.8   | 57.2      | 55.6   | **89.1**   | 57.9  |
> | GraphSearch-F  | **65.2**     | **53.3**   | 60.4      | 60.6   | 89.6   | 59.8  |
> | GraphSearch-R  | 62.6     | 48.7   | **61.0**      | **61.6**   | 87.8   | **64.6**  |
>
> Results on node classification show that **GraphSearch consistently outperforms the strong baseline Search-o1** with a smaller-size backbone, further validating the effectiveness of our structure-aware agentic framework even under limited model capacity.
>
> We have updated the evaluation in Appendix *A.2.3*.

---

### Official Review · Reviewer_RAhu · 2025-11-01

**Soundness:** 3
**Presentation:** 4
**Contribution:** 3
**Rating:** 4
**Confidence:** 4

**Summary:**

GraphSearch extends the idea of "search-augmented reasoning" to graph-structured data, aiming to solve graph learning tasks such as node classification and link prediction in zero-shot scenarios without task-specific fine-tuning.
GraphSearch use a Graph-aware Planner to disentangle search space and a Graph-aware Retriever to construct candidate sets and rank candidates based on a hybrid scoring function.

**Strengths:**

1. **The Combination of Large Reasoning Model and Graph-Aware Reasoning is Illuminating**: GraphSearch use LRM to guide graph search process by generating graph-search instructions, which can enable expressive search space control. And equip LRM with enabling zero-shot graph learning
2. **Detailed Analysis of Comprehensive Experiments**: This work provides a thorough empirical evaluation across six diverse benchmarks for both node classification and link prediction tasks across multiple graph domains. Comparing against strong baselines, this work demonstrates good performance of GraphSearch.
3. **Exemplary Clarity and Presentation**: The paper is well-written and structured, significantly enhancing its accessibility. The authors effectively use visual aids and a clear flow to articulate a complex framework, making it easy for the reader to grasp the core innovations and their motivations.

**Weaknesses:**

1. **Low Computational Efficiency and Limited Performance Improvement**: All the target tasks to be addressed in this paper, which are node classification and link prediction tasks, are not complex enough to require the introduction of LRM for completion. Introducing LRM for reasoning will significantly increase the time cost of reasoning, while there is no significant improvement in performance compared with methods based on Graph Neural Networks such as GCN and GraphPrompter.
2. **Missing Related Work about Search-Augmented Graph Learning**: Currently, Some search-augmented frameworks for graph data have already been investigated. For example, RAGraph[1] introduces RAG into graph learning, which is similar with the insight of this paper. It's better to incorporate these related works.
3. **Highly Relies on the Ability of LRM**: Using LRM to generate graph-search instructions places high demands on the capabilities of LRM itself. In the paper, merely employing *Qwen2.5-32B-Instruct* and *Qwen2.5-72B-Instruct-AWQ* models that lack long reasoning capabilities during their own pre-training process is not sufficiently convincing. It is recommended to add experiments with models that possess long reasoning capabilities (such as DeepSeek-R1) and models with smaller parameter sizes to enhance the persuasiveness of the research.

[1] RAGraph: A General Retrieval-Augmented Graph Learning Framework

**Questions:**

1. Will the attribute of graph information provided in the <information>...</information> lose topological context for the graph data? Since the information in  text modal is very limited.

---

> ### Author Response · Authors · 2025-12-03
>
> **W1**: *Low Computational Efficiency and Limited Performance Improvement*
>
> **R1**: **We respectfully clarify that node classification and link prediction are far from trivial.**
> They require multi-hop aggregation, structural reasoning, and fine-grained semantic discrimination. This is evident from the poor performance of even strong LLMs (e.g., 72B Qwen2.5-Instruct) under zero-shot CoT, suggesting that the difficulty lies in graph reasoning itself.
>
> While **GNNs** are standard methods for these tasks, they **suffer from poor transferability and rely heavily on labeled data**, which leads to high annotation costs and hampers scalability to new domains. This motivates GraphSearch, an agentic, training-free framework designed to generalize without retraining. To ensure **fair zero-shot evaluation**, we adopt a standardized protocol where models are trained on one graph set (Cora/PubMed) and evaluated on a disjoint target set (Products/Sports/Computers), with no overlap. Under this setting for node classification tasks (%), our training-free, agentic framework generalizes effectively across graphs without retraining.
>
> | Model           | Products | Sports | Computers |
> |-----------------|----------|--------|-----------|
> | GCN             | 2.89     | 14.78  | 16.56     |
> | GraphPrompter   | 15.42    | 9.26   | 26.40     |
> | GraphSearch-R | **70.8** | **61.9** | **68.1** |
>
> Our training-free agentic framework outperforms GCN and GraphPrompter, highlighting the practical value of LLM-based reasoning in label-scarce scenarios. We have updated the *Introduction* with clear motivation and included these results in the revised *Appendix A.2.2*.
>
>
>
>
>
> **W2**: *Missing Related Work about Search-Augmented Graph Learning*
>
> **R2**: We thank the reviewer for pointing out RAGraph, and we have included it in the related work section. Although both approaches employ retrieval on graph data, they follow **different paradigms**:
>
> - RAGraph is a **GNN-based, one-time static retrieval** method: it augments GNN inference by retrieving pre-constructed subgraph embeddings derived from pre-trained GNNs.
> - GraphSearch, in contrast, is an **agentic, training-free** framework that performs **multi-step and on-demand retrieval** guided by LRM reasoning, without requiring GNNs, precomputed subgraphs, or supervised training.
>
> We have updated the *Related Work* section accordingly.
>
>
>
>
>
> **W3**: *Highly Relies on the Ability of LRM*
>
> **R3**: GraphSearch is a methodological framework rather than a model-specific benchmark, and it **does not rely on particular LRMs**. We conduct experiments with different backbones, including a smaller **Qwen2.5-7B-Instruct** model and the **close-source DeepSeek-R1**, to further show that the agentic search mechanism works robustly across different LRM capacities.
>
> *Backbone: Qwen2.5-7B-Instruct*
>
> | Model           | Products | Sports | Computers | Reddit | PubMed | Cora |
> |-----------------|----------|--------|-----------|--------|--------|-------|
> | Search-o1       | **65.3** | 47.8   | 57.2      | 55.6   | 89.1 | 57.9  |
> | GraphSearch-R   | 65.2     | **53.3** | **60.4**    | 60.6  | **89.6** | 59.8  |
> | GraphSearch-F   | 62.6     | 48.7   | 61.0      | **61.6**  | 89.3   | **64.6** |
>
> *Backbone: DeepSeek-R1*
>
> | Model           | Products | Sports | Computers | Reddit | PubMed | Cora |
> |-----------------|----------|--------|-----------|--------|--------|-------|
> | Search-o1       | 73.0     | 79.3   | 64.9      | 74.7   | **90.3** | 69.0  |
> | GraphSearch-R   | 74.3     | 80.1   | 74.9      | 80.7   | 89.8   | **72.1** |
> | GraphSearch-F   | **75.6** | **80.2** | **75.1**    | **81.7**  | 89.5   | 72.0  |
>
> Overall, GraphSearch is not dependent on high-capability LRMs; rather, it provides a general agentic procedure that **consistently improves reasoning in graph learning tasks** with different-sized backbone reasoning models.
>
> We have updated the robustness experiments with different backbone sizes in *Appendix A.2.5*.
>
>
> **Q1**: *Does Text Input Lose Graph Topology?*
>
> **R4**: We believe “[object Object]” is a rendering typo referring to the <information> field. The information passed to the LLM does **not** lose graph topology. In GraphSearch, topological context is injected during reasoning:
>
> - **Structure-aware retrieval**: neighbors are selected via specific hop values, so retrieved information is already filtered by structural relevance.
> - **Edge semantics in prompts**: the planner prompt explains the meaning of edges (e.g., *cited by*, *co-purchased with*), enabling the LLM to interpret relationships consistent with the graph structure.
>
> Thus, the textual input to the LLM remains **topology-aware** rather than raw or isolated attributes.

---

### Author Response · Authors · 2025-12-03

Dear reviewers and AC,

We sincerely thank you for the thoughtful feedback. Following the rebuttal, we clarified all concerns, conducted additional experiments, and uploaded a revised version of the paper.

During the rebuttal, we have
**(i)** **clarified the misunderstanding of the relationship between our method and GraphRAGs** from reviewers ***yBUY*** and ***VWwh***. We emphasized that these approaches target different tasks and construct graph priors differently, and included GraphRAG baselines to show our performance gains;
**(ii)** **addressed the robustness concerns** raised by reviewers ***RAhu***, ***yBUY***, and ***VWwh*** by incorporating three additional backbone models, ranging from smaller to stronger reasoning models, highlighting the stability of our approach;
**(iii)** **responded to concerns about adaptation to trainable regimes** raised by reviewers ***yBUy*** and ***VWwh*** by implementing an RL-Tuned variant using GRPO, which achieves additional performance gains under supervision, demonstrating the extensibility of our method.


Our key updates are summarized below:

**1. Clearer positioning and more precise contributions:**
- **Clarified the difference with GraphRAGs.** We clarify that our work introduces graph-guided agentic reasoning for graph learning tasks. Unlike GraphRAG and its variants, which build synthetic co-occurrence graphs over text chunks, our method operates directly on real graph topology and addresses standard graph learning tasks. While GraphCoT also leverages native graph links, it is tailored to factoid QA with fixed function calls and lacks graph-aware query generation, fundamentally differing from our goal of discriminative graph prediction. We included these works for clearer positioning. ***(Sec1 Introduction, Sec2 Related work)***

- **Clarified “first agentic framework for graph learning” phrasing.** We refine the claim to avoid overstatement: our contribution is not proposing a new agent paradigm, but adapting agentic reasoning to graph learning for the first time. This is particularly valuable in training-free scenarios and enables multi-hop, structure-informed retrieval uniquely suited to graph tasks. We rephrased these statements for clearer positioning. ***(Abstract, Sec1 Introduction, Sec2 Related work)***


**2. Broader baseline comparison:**
- **Graph Learning Methods:** We conduct zero-shot evaluation across two major categories: GNNs (GCN, GraphSAGE) and GraphLLMs (GraphPrompter). Results support the motivation for our training-free agentic paradigm and demonstrate the strong transferability and effectiveness of GraphSearch. ***(Sec4.2, Appendix A.2.2)***

- **GraphRAGs:** We extend baselines to include RAG-style methods (GraphRAG, HippoRAG2, Graph-COT). Despite using stronger backbones, these models are consistently outperformed by our graph-guided agentic reasoning on standard graph learning tasks. ***(Sec4.1, Sec4.2, Appendix A.2.5)***


**3. More solid obligation study:** We significantly expanded experiments to strengthen the evidence base:

- **Impact of various backbones:** We evaluate GraphSearch across a range of LLM backbones, from the smaller Qwen2.5-7B-Instruct to the larger LLaMA-3.3-70B and the stronger open-source DeepSeek-R1. Results demonstrate consistent performance gains over strong baselines, confirming the robustness of our method across model capacities. ***(Sec4.3, Appendix A.2.3)***

- **Add one more variant: RL Trained version:** We include a learnable variant of GraphSearch trained using a lightweight GRPO-based RL approach with a simple reward design. This demonstrates the framework’s easy adaptation to supervised settings and yields further performance gains. ***(Sec4.4, Appendix A.2.4)***

- **More efficiency analysis with breakdown:** We report total token usage and its distribution across reasoning phases (Think, Search, Information, Answer), showing that GraphSearch matches the strong baseline in cost without introducing additional overhead. ***(Sec4.5, Appendix A.2.6)***


We hope these clarifications and substantial additions further strengthen the contributions and empirical support of the paper. ****Thank you again for the thoughtful feedback and the opportunity to improve our work.****

---

### Meta-Review · Area_Chair_FDG1 · 2026-01-05

**Summary:**

While the paper presents a well-written and carefully engineered framework for applying search-augmented, agentic reasoning to graph-structured data, the reviewers’ concerns ultimately outweigh its strengths. The core methodology is widely viewed as an adaptation of existing agentic RAG and search-based reasoning paradigms rather than a fundamentally new technical contribution. Key components such as the planner, retrieval scopes, and hybrid scoring are largely policy- and prompt-driven, with limited algorithmic novelty or theoretical grounding. The framework relies substantially on large reasoning models and prompt-based design choices, which raises some open questions about generality and principled formulation. While the paper is well executed and empirically strong, it does not yet provide a theoretical analysis, a learning-based core formulation, or clearly novel retrieval or reasoning mechanisms beyond existing agentic search paradigms. As a result, despite its solid experimental performance, the contribution may be viewed as incremental relative to the acceptance bar.

**Reviewer Concerns:**

After the rebuttal, the following points might have been addressed:
- Efficiency and necessity of LRMs (RAhu, yBUY): Addressed with detailed token/latency breakdowns showing comparable cost to Search-o1, and strong zero-shot gains over GNNs.
- Reliance on LRM capability (RAhu, yBUY): Addressed via experiments with varied backbones showing consistent gains.
- Missing baselines (RAhu, yBUY, VWwh): Addressed through expanded discussion and empirical comparisons with RAGraph, GraphRAG, HippoRAG2, Graph-CoT, and additional GNNs.

Remaining concerns include:
- Limited fundamental novelty / policy-driven design (yBUY, LMYp, VWwh): While the authors clarify positioning and motivate the graph-specific components, some reviewers may still view the contribution as primarily an adaptation rather than a deep algorithmic advance.
- Lack of theoretical analysis (LMYp): Acknowledged but deferred to future work.
- Heavy reliance on prompt templates and lack of learning (LMYp, VWwh): Mitigated by some additional experiments, but the core method remains largely heuristic.
- Reproducibility (VWwh): Promises of code release help, but no public code yet.

**Reviewer Scores:**

Overall, the reviewers' major concerns persist as described above; therefore, I believe the decision might not have been changed if the reviewers had had a chance to participate in the discussion.

---

### Decision · Program_Chairs · 2026-01-26

Reject